# Assessment of Nutritional Status in Children with Familial Mediterranean Fever Using Prognostic Nutritional Index and Controlling Nutritional Status Score: Relationship with Clinical Findings and MEFV Mutations

**DOI:** 10.3390/children12111540

**Published:** 2025-11-14

**Authors:** Seyda Dogantan, Adem Keskin, Burcu Bozkaya Yücel, Peren Perk, Emel Hatun Aytaç Kaplan, Rahime Koç, Sanem Eren Akarcan

**Affiliations:** 1Department of Pediatric Rheumatology, Basaksehir Cam and Sakura City Hospital, Istanbul 34480, Turkey; rahime-koc@hotmail.com; 2Department of Medical Biochemistry, Faculty of Medicine, Aydin Adnan Menderes University, Aydin 09100, Turkey; adem.keskin@adu.edu.tr; 3Department of Pediatric Rheumatology, Samsun Research and Education Hospital, Samsun 55090, Turkey; bozkayaburcu@hotmail.com; 4Department of Pediatric Neurology, Basaksehir Cam and Sakura City Hospital, Istanbul 34480, Turkey; perkperen@gmail.com; 5Department of Pediatric Endocrinology, Basaksehir Çam ve Sakura City Hospital, Istanbul 34480, Turkey; emel_ctf@hotmail.com; 6Department of Pediatric Immunology and Allergy, Izmir City Hospital, Izmır 35530, Turkey; saneren@yahoo.com

**Keywords:** familial Mediterranean fever, controlling nutritional status score, Mediterranean fever gene, prognostic nutritional index

## Abstract

**Highlights:**

**What are the main findings?**
The distribution of CONUT scores by subcategory differs between kids with FMF and healthy kids.Higher total cholesterol levels in children with FMF are associated with an increased number of symptom days.

**What are the implications of the main findings?**
Routine monitoring of total cholesterol levels in children with FMF may reduce the risk of complications such as amyloidosis and persistent malnutrition.Routine monitoring of total cholesterol levels in children with FMF may reduce the number of symptom days.

**Abstract:**

**Background/Objectives**: The effect of nutritional status on the prognosis of Familial Mediterranean Fever (FMF), a hereditary autoinflammatory illness, is unclear. This research aims to investigate whether nutritional status indicators, such as the Controlling Nutritional Status (CONUT) score and the Prognostic Nutritional Index (PNI), differ in kids with FMF compared to healthy kids. It also aims to investigate the possible relation between these indicators and the types of *MEFV* gene mutations detected in kids with FMF. **Methods**: The research included 90 kids with FMF and 90 healthy children as controls. The FMF group was further divided into three subgroups based on MEFV gene mutation status. The PNI and CONUT scores of these groups and subgroups were compared. **Results**: A difference was found in the distribution of CONUT scores in the FMF group compared to the healthy group. However, there was no difference in the distribution of PNI between the two groups. C-reactive protein, triglyceride, and total cholesterol values were higher in the FMF group than in the control group. A difference was also determined between the two groups in the distribution of total cholesterol scores categorized by CONUT score. A negative correlation was found between this categorized score and the number of symptom days. No significant difference was found in the distribution of PNI and CONUT scores among subgroups based on MEFV gene mutation status. **Conclusions**: In children with FMF, total cholesterol levels should be routinely monitored longitudinally, even if they remain within reference values, to prevent some complications in adulthood.

## 1. Introduction

Familial Mediterranean Fever (FMF), the most global monogenic autoinflammatory illness worldwide, is characterized by recurrent stereotypic febrile attacks, usually lasting less than 3 days, and primarily affects children of Mediterranean descent. These attacks are often accompanied by acute abdominal pain and biological inflammatory syndrome. Diagnosis is usually confirmed by mutations in the Mediterranean Fever (MEFV) gene, particularly in exon 10 [1]. Although FMF is considered an episodic disease characterized by short-lived attacks, various chronic inflammatory conditions associated with the disease have also been identified. Colchicine is the mainstay of FMF treatment, and interleukin-1 antagonists are the preferred treatment option in cases of colchicine resistance or intolerance [2]. Furthermore, FMF, a hereditary disease of unknown cause, is a condition where the role of specific foods as triggers is being investigated, and conflicting results have been reported regarding the recurrence of FMF attacks with the consumption of fatty and salty foods. On the other hand, a diet rich in antioxidants and anti-inflammatory food supplements have been reported to partially reduce symptoms and improve well-being in patients with FMF [3].

The Controlling Nutritional Status (CONUT) score was first proposed in 2005 as a simple and practical tool for assessing and monitoring the nutritional status of hospitalized patients. To obtain this score, total cholesterol, serum albumin, and lymphocyte levels are categorized and scored within specific ranges, and the total score is the sum of these scores [4,5]. Based on these three biochemical parameters, the CONUT score reflects serum albumin protein reserves and liver function; total cholesterol reflects fat metabolism and calorie consumption; and the lymphocyte count reflects the status of the immune system [6]. The CONUT score has been proposed as a promising prognostic indicator in a wide variety of scenarios and clinical situations, from cancer and sepsis to diabetes and cardiovascular disease, from dialysis patients to stroke patients, and from organ transplant patients to those undergoing chemotherapy [6,7,8,9,10,11,12,13].

The Prognostic Nutritional Index (PNI) is an indicator that holistically assesses an individual’s immune and nutritional status. It is calculated by multiplying the serum albumin level and total lymphocyte count by specific coefficients and then summing these values [14]. Like the CONUT score, the PNI has been the subject of numerous studies in recent years, and its relationship with prognosis in various diseases has been investigated. In this context, a low PNI has been reported to be correlated with a poor prognosis and a higher risk of complications, while a high PNI indicates a better prognosis [14,15,16,17,18,19,20].

The relationship between FMF, a hereditary autoinflammatory disease, and nutritional status is a controversial issue. CONUT and PNI markers are important indicators for assessing nutritional status and are correlated with the prognosis of many diseases. This research aims to investigate whether these indicators differ in children with FMF compared to healthy children. It also aims to examine the relationship between these differences and the presence of symptoms and the number of days they persist. It also aims to evaluate the biochemical parameters associated with these indicators in children with FMF and to investigate the possible relationship between MEFV gene mutation types and these indicators.

## 2. Materials and Methods

### 2.1. Research Population

The number of kids involved in the research was determined using the G*Power 3.1.9.7 power analysis program (Test family: *t* tests, Statistical test: Means: Difference between two independent means (two groups), Type of power analysis: A priori: Compute required sample size-given α, power, and effect size, (Effect size d = 0.5, α = 0.05, power (1 − β) = 0.95, Allocation ratio N2/N1). As a result of the power analysis, it was determined that the minimum number of kids required for each group was 88. Therefore, ninety patients diagnosed with FMF who were followed in the Rheumatology Clinic of the Department of Child Health and Diseases at Basaksehir Cam and Sakura Hospital in Istanbul were included in the study as the FMF group. FMF diagnoses were confirmed according to the Eurofever/Pediatric Rheumatology International Research Organization (PRINTO) criteria [21]. In addition, the FMF diagnosis was based on a combination of Tel-Hashomer clinical criteria and genetic analysis [22]. All patients were receiving colchicine and other anti-inflammatory or anticoagulant medications. Participants younger than 2 years or older than 18 years, those with amyloidosis or proteinuria, those with any chronic comorbidities, particularly those with concurrent hematologic disorders, those receiving corticosteroid therapy, and those with active infections were excluded from study. Single heterozygous patients carrying a variant of uncertain significance (VUS) in the MEFV gene were also excluded from the research. Additionally, ninety healthy children without malignancy, chronic disease, inflammatory or hematological disorders, or medication use were included in the study as the control group.

### 2.2. CONUT Score Assessment

The CONUT score is a score obtained by categorizing the total lymphocyte count, serum albumin, and total cholesterol values. Specifically, if the total lymphocyte count is >1600/mm^3^, 0 points are given; if it is between 1200–1599/mm^3^, 1 point is given; if it is between 800–1199/mm^3^, 2 points are given; and if it is <800/mm^3^, 3 points are given. If the serum albumin level is ≥3.5 g/dL, 0 points are given; if it is 3.0–3.49 g/dL, 2 points are given; if it is 2.5–2.99 g/dL, 4 points are given; if it is <2.5 g/dL, 6 points are given. If the total cholesterol level is >180 mg/dL, 0 points are given; if it is 140–180 mg/dL, 1 point is given; if it is 100–139 mg/dL, 2 points are given; if it is <100 mg/dL, 3 points are given. Nutritional status is determined based on the total score obtained from these three parameters: 0–1 points is classified as normal nutrition, 2–4 points is mild malnutrition, 5–8 points is moderate malnutrition, and 9–12 points is severe malnutrition [4,5].

### 2.3. PNI Score Assessment

PNI is an index obtained by multiplying the total lymphocyte count and serum albumin levels by specific coefficients and summing them. Accordingly, PNI is calculated as the sum of 0.005 times the total lymphocyte count per 1 mm^3^ of blood and 10 times the albumin level measured in grams per 1 dL of serum. A PNI value > 45 indicates normal nutritional status (Category 1), 40–45 indicates mild malnutrition (Category 2), and <40 indicates moderate-to-severe malnutrition with a poor prognosis (Category 3) [14].

### 2.4. Statistical Analysis

SPSS 22 for Windows (IBM, New York, NY, USA) was used for statistical data analysis. The conformity of continuous variables to normal distribution was assessed using the Shapiro–Wilk test using skewness and kurtosis values. Continuous variables are shown as mean ± standard deviation (X ± SD), and categorical variables are shown as *n* (percentage frequency). The independent samples t-test and Oneway-Anova test (Post Hoc Dunnet T3 test) were used among parametric tests to compare continuous variables. The Kruskal–Wallis test and Mann–Whitney U test were used among parametric tests to compare continuous variables that did not show a normal distribution. The chi-square test was used to compare categorical variables. *p* < 0.05 was accepted as the limit of statistical significance.

## 3. Results

The research involved 90 patients diagnosed with FMF, aged between 2 and 17 years. The control group consisted of 90 healthy kids aged between 1 and 16 years. Descriptive information for groups is presented in Table 1.

No significant difference was found between the two groups in terms of gender ratios and Body Mass Index (*p* = 0.55, *p* = 0.42, respectively) (Table 1). The mean age, height, and weight of the FMF group were found to be higher than the values of the control group (*p* = 0.001, *p* = 0.022, *p* = 0.026, respectively) (Table 1). A family history of FMF was found in 26.67% of patients, and a history of consanguinity was found in 23.33% (Table 1). Furthermore, the most frequently reported symptom was abdominal pain (82.22%) (Table 1). Furthermore, the most common symptom duration was 3 days (44.44%) (Table 1). Laboratory results for the FMF and control groups are presented in Table 2.

In the FMF group, mean CRP, triglyceride and BUN levels were above the reference ranges. Other laboratory findings remained within the reference range. Meanwhile, mean serum amyloid A levels were within the reference range, but the distribution of values was abnormal. Mean vitamin D levels were also within the reference range but below optimal values (>30 ng/mL) [23].

CRP, triglyceride, and total cholesterol values were higher in the FMF group than in the control group (Table 2). No significant difference was found between the two groups in terms of ESR, albumin, and lymphocyte levels (Table 2).

The PNI scores of children in both groups were above 45, meaning their PNI scores were classified as Category 1. Additionally, serum albumin levels, total lymphocyte counts, and total cholesterol levels were categorized to determine the CONUT score in both groups. The data obtained in this context are presented in Table 3.

While no important difference was found between the sub categorical distribution of the groups based on albumin and total lymphocyte counts, a significant difference was found in the sub categorical distribution of total cholesterol levels and CONUT scores distribution (Table 3).

Correlation analysis was performed to examine the correlation between FMF group CONUT scores, the categorical scores of total cholesterol levels, and the presence of symptoms and the number of symptom days. The analysis revealed a negative correlation between the categorical scores of total cholesterol levels and the number of symptom days (correlation coefficient = −0.247, *p* = 0.019).

A total of 9 *MEFV* gene mutations were detected in kids in the FMF group, 6 of which were pathogenic. Fifteen of these patients (16.67%) were homozygous, 45 (50%) were heterozygous, and 30 (33.33%) were compound heterozygous. The distribution of *MEFV* gene mutations in the FMF group is presented in Table 4.

In the distribution of MEFV gene mutations in the FMF group, the M694V mutation was detected at the highest rate with 66.67% (Table 4). The FMF group was divided into three subgroups according to the mutation status in the *MEFV* gene: homozygous, heterozygous, and compound heterozygous. The average age of the homozygous subgroup was 11.20 ± 4.60, the average age of the heterozygous subgroup was 9.98 ± 3.78, and the average age of the compound heterozygous group was 11.53 ± 3.94. In the homozygous subgroup, 60% (*n* = 9) were female and 40% (*n* = 6) were male. In the heterozygous group, these rates were 42.22% (*n* = 19) and 52.78% (*n* = 26), respectively. In the compound heterozygous group, these rates were 53.33% (*n* = 16) and 46.67% (*n* = 14), respectively. No important difference was found in terms of gender ratios or average ages among the three subgroups. The symptoms, CONUT and total cholesterol subcategory distributions and mean CRP levels for these three subgroups are presented in Table 5.

An important difference was found in the incidence of joint pain among the homozygous, heterozygous, and combined heterozygous subgroups for the MEFV gene mutation (Table 5). The incidence of joint pain symptoms was found to be lower in the homozygous subgroup than in the heterozygous and combined heterozygous subgroups (*p* = 0.004 and *p* = 0.020, respectively). No important differences were found in the incidence of other symptoms, the distribution of CONUT scores by subcategories, the distribution of total cholesterol by subcategories, the mean of CRP levels, and the mean of symptom-day (Table 5).

## 4. Discussion

The aim of this study was to investigate whether indicators such as PNI and CONUT, developed to assess nutritional status in children with FMF, differ from those in healthy children, and to examine the possible relationship between these indicators and clinical findings and MEFV gene mutations. The study revealed that the distribution of CONUT scores in children with FMF differed from that in healthy children according to the subcategory, but PNI values remained within normal limits in both groups. Furthermore, cholesterol levels associated with CONUT scores were higher in children with FMF than in healthy children. This indicates that serum albumin and lymphocyte levels are generally preserved in children with FMF, but differences in total cholesterol levels increase the CONUT score. Moreover, the negative correlation between total cholesterol categorical scores and symptom days suggests that high cholesterol levels may be associated with disease activity. Additionally, triglyceride levels, another lipid metabolism parameter, were higher in children with FMF compared to healthy children, parallel to total cholesterol levels. There is limited literature on the relationship between nutritional status, lipid metabolism, and disease activity in FMF. The current study is important as it demonstrates this relationship in the pediatric population for the first time. The findings suggest that total cholesterol levels and CONUT scores should be monitored regularly from childhood onwards.

Malnutrition is a broad concept encompassing both undernutrition and overnutrition, and it is quite common in pediatric FMF patients. Malnutrition is a significant clinical problem in children with FMF, and this condition can often be corrected with colchicine treatment. However, the presence of malnutrition at diagnosis and more severe phenotypes of the disease may be major risk factors for the development of persistent malnutrition in the future. These findings indicate that FMF is not limited to inflammatory attacks but can also negatively affect children’s growth and development process and overall nutritional status [24]. However, a recent study examining data from seven studies examining the relationship between diet and the clinical course of FMF reported that there is no definitive evidence for the effect of diet in triggering FMF symptoms, and further researchers are needed to clarify this hypothetical relationship [3]. On the other hand, PNI and CONUT scores, used to assess nutritional status, have recently been investigated as predictors of illness activity and underlying disease prognosis in various clinical settings. In this context, a recent study in adult FMF patients reported that these indicators can be used to predict the development of amyloidosis in FMF individuals. They also reported that a high CONUT score, in addition to age and the M694V homozygous mutation, was associated with the development of amyloidosis in FMF individuals [25].

In this study comparing the results of PNI and CONUT scores used to assess nutritional status in children with FMF with those of healthy children, it was found that there was a difference in the distribution of CONUT scores across subcategories between children with FMF and healthy children, but PNI remained within normal limits in both groups. This indicates that, despite the general preservation of albumin and lymphocyte values in children with FMF, differences in total cholesterol levels elevate the CONUT score.

In a recent study involving 266 children with FMF, high-fat food consumption was reported as a trigger for attacks in 15.8% of patients. In contrast, 8.4% of patients reported that a low-fat diet was helpful in reducing the frequency of attacks and became one of the nutritional management strategies they adopted [26]. Another recent study reported that adult patients with FMF had higher total cholesterol levels compared to healthy individuals. Moreover, these patients’ total cholesterol levels were also above reference values [27]. A study comparing adult FMF individuals without and with AA amyloidosis reported that total cholesterol levels were above the reference range in both groups, but there was no important difference between the groups. On the other hand, triglyceride values were reported to be higher in FMF individuals with amyloidosis compared to those without amyloidosis [28]. Similarly, another recent study reported that triglyceride values were higher in children with FMF than in healthy children [29].

In this study evaluating children with FMF, although total cholesterol and serum amyloid A levels were within reference ranges, total cholesterol levels were higher than in healthy children. Triglyceride levels were also higher in children with FMF and slightly above reference values. Furthermore, total cholesterol categorical scores based on the CONUT score were not related to the presence of any symptoms but were found to be negatively correlated with the number of symptom days. In other words, as total cholesterol categorical scores decrease, cholesterol levels increase, and the number of symptom days increases accordingly. This finding suggests that high total cholesterol levels may be associated with the duration of FMF symptoms. In addition, the fact that cholesterol levels remained within the reference range despite being high may be due to the fact that the research was conducted in a pediatric population.

This study also created homozygous, heterozygous, and compound heterozygous subgroups based on MEFV gene mutation status. No important differences were determined between these subgroups in terms of PNI and CONUT scores. Similarly, no significant difference was found between the subgroups in terms of the number of symptom days, the presence of symptoms other than joint pain, CRP level, PNI and CONUT score subcategories. In contrast, compared with healthy children, children with FMF had higher CRP levels and suboptimal vitamin D levels.

This research is the first to evaluate the nutritional status of children with FMF based on CONUT scores and PNI, according to our literature review, and this aspect constitutes an important strength. However, the fact that the findings were not supported by different parameters or methods associated with different lipid metabolism can be considered as one of the limitations of this research. The age difference between the two groups can also be considered a limitation of the study. However, the lack of a statistically important difference in body mass index and gender ratio between the two groups suggests that the groups were comparable in terms of overall physical condition. On the other hand, it is noteworthy that total cholesterol levels in children with FMF, although within the reference range, are higher than in healthy children and are related to the number of symptom days. It is also noteworthy that triglyceride levels associated with lipid metabolism were found to be above reference values and higher than in healthy children. Indeed, considering that total cholesterol levels in adult FMF patients are mostly above reference values, it is clear that lipid metabolism in FMF needs to be investigated at a more advanced level. The potential of our study to serve as a reference for future studies in this direction is another strength of our research. Furthermore, considering the study’s findings, the CONUT score can be used as a practical tool for assessing nutritional status in children with FMF. Because changes in total cholesterol levels can be reflected in the CONUT score, this score can aid in monitoring nutritional status during clinical follow-up and early identification of potential malnutrition risks. Furthermore, given the correlation of total cholesterol levels associated with the CONUT score with FMF symptom duration and disease activity, it can guide nutritional intervention planning to prevent long-term complications (e.g., amyloidosis, chronic malnutrition).

## 5. Conclusions

CONUT scores and total cholesterol and triglyceride levels are higher in kids with FMF compared to healthy kids. While higher CONUT scores are observed, the reason for no difference in PNI between kids with FMF and healthy kids is the categorical scoring of total cholesterol levels used in the CONUT score calculation. Furthermore, this categorical scoring is inversely proportional to the number of symptom days. In children with FMF, total cholesterol levels should be monitored throughout childhood to prevent persistent malnutrition, amyloidosis, and increases in total cholesterol levels above reference values over time.

## Figures and Tables

**Table 1 children-12-01540-t001:** Descriptive information of groups.

Descriptive Information		FMF Group (*n* = 90)		Control Group (*n* = 90)
Age X ± SD		10.70 ± 4.00		8.82 ± 3.73
Gender *n* (%)	Female	44 (48.89)		48 (53.33)
	Male	46 (51.11)		42 (46.67)
Height (cm) X ± SD		140.53 ± 22.00		133.11 ± 21.15
Weight (kg) X ± SD		39.01 ± 17.54		33.48 ± 15.53
BMI (kg/m^2^) X ± SD		18.61 ± 3.75		18.13 ± 4.27
Age at disease onset (months) X ± SD		41.63 ± 48.00		-
Family history of FMF *n* (%)		24 (26.67)		-
Consanguineous marriage *n* (%)		21 (23.33)		-
Family history of rheumatic disease *n* (%)		14 (15.56)		-
Family history of rheumatic disease *n* (%)		Rheumatoid arthritis	13 (14.44)	-
		Takayasu	1 (1.11)	-
Symptom *n* (%)		Abdominal pain	74 (82.22)	-
		Joint pain	54 (60)	-
		Chest pain	30 (33.33)	-
		Fever	69 (76.67)	-
		Headache	11 (12.22)	-
		Nausea	9 (10)	-
		Diarrhea	6 (6.67)	-
Number of symptom days *n* (%)		1	11 (12.22)	-
		2	31 (34.44)	-
		3	40 (44.44)	-
		4	5 (5.56)	-
		7	3 (3.33)	-
Mean symptom days X ± SD		2.60 ± 1.13		

FMF: Familial Mediterranean Fever, BMI: Body Mass Index, X ± SD: Mean ± Standard Deviation.

**Table 2 children-12-01540-t002:** Laboratory results of the groups.

Parameters	Reference Range	FMF Group (*n* = 90)	Control Group (*n* = 90)	*p*
ESR	0–20 mm/h	9.87 ± 8.41	10.78 ± 11.37	0.540
CRP	<5 mg/L	7.83 ± 23.14	2.16 ± 4.18	0.024
Albumin	3.5–5.0 g/dL	4.59 ± 0.28	4.63 ± 0.26	0.322
Lymphocyte	1.000–4.800/mm^3^	2907 ± 925.08	3073.44 ± 1177.08	0.294
Total cholesterol	<170 mg/dL	143.87 ± 29.76	126.77 ± 22.04	<0.001
Triglyceride	<90 mg/dL	102.64 ± 21.33	83.58 ± 14.82	<0.001
WBC	5–14 × 10^3^/µL	7.64 ± 2.09	7.19 ± 1.93	0.136
Hematocrit	35–45%	38.6 ± 5.37	37.5 ± 4.77	0.148
Hemoglobin	11–16 g/dL	12.64 ± 1.57	12.88 ± 0.94	0.217
ANC	1.5–8 × 10^3^/µL	3.98 ± 1.72	3.88 ± 2.58	0.760
ALC	1–4 × 10^3^/µL	2.86 ± 0.91	2.98 ± 0.78	0.344
Platelet	150–400 × 10^3^/µL	328.27 ± 81.27	312.55 ± 95.43	0.236
MCV	75–95 fL	79.3 ± 4.81	79.28 ± 4.24	0.237
Serum Amyloid A	<10 mg/L	6.11 ± 20.89	-	
Vitamin B12	200–900 pg/mL	394.82 ± 213.54	-	
Vitamin D	20–50 ng/mL	21.5 ± 8.59	-	
Folic acid	3–17 ng/mL	7.25 ± 3.8	-	
BUN	7–20 mg/dL	22.99 ± 6.08	-	
Creatinine	0.3–0.8 mg/dL	0.54 ± 0.17	-	
ALT	5–35 U/L	15.93 ± 7.71	-	
AST	5–35 U/L	23 ± 6.29	-	
LDH	100–250 U/L	216.33 ± 41.71	-	
Total Protein	60–80 g/L	73.06 ± 4.49	-	
Thyroxine	0.8–2 ng/dL	1.42 ± 0.32	-	
TSH	0.4–4.5 mIU/L	2.33 ± 0.98	-	
Calcium	8.5–10.5 mg/dL	9.71 ± 0.4	-	
Magnesium	1.5–2.5 mg/dL	2.07 ± 0.16	-	

FMF: Familial Mediterranean Fever, ESR: Erythrocyte Sedimentation Rate, CRP: C-Reactive Protein, ANC: Absolute Neutrophil Count, WBC: White Blood Cell, ALC: Absolute Lymphocyte Count, BUN: Blood Urea Nitrogen, AST: Aspartate Aminotransferase, MCV: Mean Corpuscular Volume, ALT: Alanine Aminotransferase, LDH: Lactate Dehydrogenase, TSH: Thyroid Stimulating Hormone.

**Table 3 children-12-01540-t003:** Distribution of data regarding CONUT scores of groups.

Parameter	Subcategory	FMF Group (*n* = 90)	Control Group (*n* = 90)	*p*
CONUT scores *n* (%)	0–1	42 (46.67)	87 (96.67)	<0.001
	2–4	48 (53.33)	3 (3.33)	
Albumin *n* (%)	>3.5 g/L	90 (100)	90 (100)	-
Lymphocyte counts *n* (%)	>1600/mm^3^	88 (97.78)	89 (98.89)	0.738
	1200–1600/mm^3^	1 (1.11)	0	
	800–1200/mm^3^	1 (1.11)	1 (1.11)	
Total cholesterol *n* (%)	>180	12 (13.33)	3 (3.33)	<0.001
	140–179	31 (34.44)	11 (12.22)	
	100–140	43 (47.78)	75 (83.33)	
	<100	4 (4.44)	1 (1.11)	

FMF: Familial Mediterranean Fever.

**Table 4 children-12-01540-t004:** Distribution of *MEFV* gene mutations in FMF patients (*n* = 90).

Variants	MEFV Gene Mutations	*n* (%) *	Mutation Type *n* (%)	
Pathogenic variants	M694V	60 (66.67)	Heterozygous	47 (78.33)
			Homozygous	13 (21.67)
	M680I	22 (24.44)	Heterozygous	22 (100)
			Homozygous	0
	M694I	4 (4.44)	Heterozygous	4 (100)
			Homozygous	0
	V726A	9 (10)	Heterozygous	9 (100)
			Homozygous	0
	A744S	5 (5.56)	Heterozygous	5 (100)
			Homozygous	0
	R761H	5 (5.56)	Heterozygous	5 (100)
			Homozygous	0
Variants of uncertain significance	E148Q	10 (11.11)	Heterozygous	8 (80)
			Homozygous	2 (20)
	R202Q	5 (5.56)	Heterozygous	5 (100)
			Homozygous	0
	P369S	2 (02.22)	Heterozygous	2 (100)
			Homozygous	0

* Because some patients had compound heterozygous mutations, these patients were included more than once in the percentile frequency of different mutation types.

**Table 5 children-12-01540-t005:** Distributions of symptoms, CONUT and total cholesterol subcategories and mean CRP levels in subgroups.

Parameters		Homozygous Subgroup (*n* = 15)	Heterozygote Subgroup (*n* = 45)	Compound Heterozygote Subgroup (*n* = 30)	*p*
C-Reactive Protein X ± SD		5.82 ± 12.47	10.26 ± 29.82	5.20 ± 14.21	0.502
Symptom *n* (%)	Abdominal pain	13 (86.67)	38 (84.44)	23 (76.67)	0.530
	Joint pain	4 (26.67)	31 (68.69)	19 (63.33)	0.014
	Chest pain	7 (46.67)	14 (31.11)	9 (30)	0.484
	Fever	10 (66.67)	36 (80)	23 (76.67)	0.572
	Headache	2 (13.33)	6 (13.33)	3 (10)	0.902
	Nausea	2 (13.33)	5 (11.11)	2 (06.67)	0.734
	Diarrhea	1 (06.67)	3 (06.67)	2 (06.67)	1.000
Mean symptom days X ± SD		2.27 ± 0.70	2.84 ± 1.33	2.40 ± 0.89	0.113
CONUT scores *n* (%)	0–1	6 (40)	24 (53.33)	15 (50)	0.670
	2–4	9 (60)	21 (46.67)	15 (50)	
Total cholesterol *n* (%)	>180	2 (13.33)	6 (13.33)	4 (13.33)	0.947
	140–179	4 (26.67)	18 (40)	9 (30)	
	100–140	8 (53.33)	19 (42.22)	16 (53.33)	
	<100	1 (06.67)	2 (4.44)	1 (3.33)	

## Data Availability

The data are not publicly available due to confidentiality or ethical restrictions. Additionally, The datasets used and/or analyzed during the current study are available from the corresponding author on reasonable request.

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
