# Peer review of "Assessment of Nutritional Status in Children with Familial Mediterranean Fever Using Prognostic Nutritional Index and Controlling Nutritional Status Score: Relationship with Clinical Findings and MEFV Mutations"

_children, 2025, doi:10.3390/children12111540_

Round 1

Reviewer 1 Report

Comments and Suggestions for Authors

This manuscript is well structured and addresses an original and clinically relevant topic on the relationship between nutritional indices (CONUT and PNI) and disease features in children with Familial Mediterranean Fever.

The study design, statistical power, and methodology are appropriate. However, the text would benefit from moderate linguistic improvement, reduction of redundancy, and clearer presentation of results. Minor corrections are needed in terminology, units, and formatting. Overall, it is a scientifically sound and well-designed paper that fits the scope of Children.

Comments on the Quality of English Language

Overall, the English is adequate but should undergo minor to moderate language editing before publication.

Author Response

Reviewer 1

This manuscript is well structured and addresses an original and clinically relevant topic on the relationship between nutritional indices (CONUT and PNI) and disease features in children with Familial Mediterranean Fever.

Comments 1:       The study design, statistical power, and methodology are appropriate. However, the text would benefit from moderate linguistic improvement, reduction of redundancy, and clearer presentation of results. Minor corrections are needed in terminology, units, and formatting. Overall, it is a scientifically sound and well-designed paper that fits the scope of Children.

Response 1:  Dear Reviewer,

First of all, thank you very much for taking the time to evaluate our manuscript and for your constructive feedback.

Based on your suggestions, linguistic improvements have been made to the text, unnecessary repetitions have been reduced, results have been made more understandable, and minor corrections have been made to terminology, units, and formatting.

Sincerely

Reviewer 2 Report

Comments and Suggestions for Authors

This manuscript by Dogantan, S, et al., put forward an assessment of nutritional status in children with FMF in relation with clinical onset, and proposed total cholesterol monitoring in the clinic to help reduce symptom days. 

The findings and the suggestions are relevant and helpful, especially in the FMF high occurrence region where this study is conducted. 

Several points that the authors should revise: 

1. When monitoring total cholesterol levels, plasma triglyceride levels should also be monitored. Triglyceride levels are closely and positively correlated with serum CRP levels and with overall pro-inflammatory status, as well as correlated with total cholesterol levels.  Monitoring both total cholesterol and triglyceride should be done together 

2. Please add control group’s n numbers, age, sex, weight, height, BMI, etc into Table 1 

3. Please add p value between the FMF and the control groups regarding each test value when the numbers are available in Table 2 

Author Response

Reviewer 2

This manuscript by Dogantan, S, et al., put forward an assessment of nutritional status in children with FMF in relation with clinical onset, and proposed total cholesterol monitoring in the clinic to help reduce symptom days. 

The findings and the suggestions are relevant and helpful, especially in the FMF high occurrence region where this study is conducted. 

Several points that the authors should revise: 

Comments 1:       

When monitoring total cholesterol levels, plasma triglyceride levels should also be monitored. Triglyceride levels are closely and positively correlated with serum CRP levels and with overall pro-inflammatory status, as well as correlated with total cholesterol levels.  Monitoring both total cholesterol and triglyceride should be done together

Response 1: 

Dear Reviewer,

First of all, thank you very much for taking the time to evaluate our manuscript and for your constructive feedback.

Taking your suggestions into account, triglyceride levels of both groups were included in the study.

Comments 2: Please add control group’s n numbers, age, sex, weight, height, BMI, etc into Table 1 

Response 2:

Necessary additions have been made to Table 1 in line with your suggestions.

Comments 3:

Please add p value between the FMF and the control groups regarding each test value when the numbers are available in Table 2

Response 3:

Necessary additions have been made to Table 2 in line with your suggestions.

Sincerely

Reviewer 3 Report

Comments and Suggestions for Authors

1.- The discussion should reflect the possible practical applications of CONUT in relation to FMF.

2.- What criteria were used to select the age of the study population?

3.- How can the absence of data in the control group in Table 2 be explained or justified?

4.- The degree of coincidence reaches 30% when the ideal would be around 20%. How can this be explained? 

Author Response

Reviewer 3

Comments 1:       

The discussion should reflect the possible practical applications of CONUT in relation to FMF.

Response 1: 

Dear Reviewer,

First of all, thank you very much for taking the time to evaluate our manuscript and for your constructive feedback.

Taking your suggestion into consideration, the following sentences have been added to the last part of the discussion section.

“…..Furthermore, considering the study's findings, the CONUT score can be used as a practical tool for assessing nutritional status in children with FMF. Because changes in total cholesterol levels can be reflected in the CONUT score, this score can aid in monitoring nutritional status during clinical follow-up and early identification of potential malnutrition risks. Furthermore, given the correlation of total cholesterol levels associated with the CONUT score with FMF symptom duration and disease activity, it can guide nutritional intervention planning to prevent long-term complications (e.g., amyloidosis, chronic malnutrition).”

Comments 2:

What criteria were used to select the age of the study population?

Response 2:

The study population consisted of randomly selected participants from an existing patient pool (patients followed with a diagnosis of FMF in our pediatric rheumatology clinic) rather than a predetermined age criterion. The following sentences have been added to the article regarding this topic.

“….The age difference between the two groups can also be considered a limitation of the study. However, the lack of a statistically important difference in body mass index and gender ratio between the two groups suggests that the groups were comparable in terms of overall physical condition.”

Comments 3:

How can the absence of data in the control group in Table 2 be explained or justified?

Response 3:

The authors received no funding for this study. Although the authors' location in a developing country and limited financial resources limited the study's ability to analyze some parameters, all baseline study parameters were analyzed across the two groups. Other parameters in the FMF group were included to provide information about the patients' overall clinical status.

Comments 4:

The degree of coincidence reaches 30% when the ideal would be around 20%. How can this be explained?

Response 4:

The rate has been reduced considering your criticism.

Sincerely